# Future climate change vulnerability of endemic island mammals

Camille Leclerc[1,2✉], Franck Courchamp [1] & Céline Bellard[1]

Despite their high vulnerability, insular ecosystems have been largely ignored in climate change assessments, and when they are investigated, studies tend to focus on exposure to threats instead of vulnerability. The present study examines climate change vulnerability of islands, focusing on endemic mammals and by 2050 (RCPs 6.0 and 8.5), using trait-based and quantitative-vulnerability frameworks that take into account exposure, sensitivity, and adaptive capacity. Our results suggest that all islands and archipelagos show a certain level of vulnerability to future climate change, that is typically more important in Pacific Ocean ones. Among the drivers of vulnerability to climate change, exposure was rarely the main one and did not explain the pattern of vulnerability. In addition, endemic mammals with long generation lengths and high dietary specializations are predicted to be the most vulnerable to climate change. Our findings highlight the importance of exploring islands vulnerability to identify the highest climate change impacts and to avoid the extinction of unique biodiversity.

[1] Université Paris-Saclay, CNRS, AgroParisTech, Ecologie Systématique Evolution, 91405 Orsay, France. [2] Present address: INRAE, Univ. of Aix Marseille, UMR RECOVER, Aix-en-Provence, France. ✉email: camille.leclerc@inrae.fr

The main drivers of current biodiversity loss include agricultural expansion, overexploitation, and introduction of invasive alien species[1], but several lines of research suggest that climate change could become a prominent, if not the leading, cause of extinction[2–4]. This is notably true for islands, which represent <5% of the earth's surface yet host unique species, mostly with singular characteristics that make them particularly prone to this threat[5]. Indeed, insular species are often insufficiently adapted to changing environments as a consequence of their isolated evolution (i.e. insularity syndrome[6,7]) and the finite nature of insular ecosystems. For instance, island conditions drastically limit the potential of species' responses to climatic shifts by moving in latitudes or altitudes due to the restricted areas of islands or migration to other landmasses, which is less likely for continental species[7]. A critical issue is thus assessing and understanding climate change impacts on insular species, particularly endemic species because their richness exceeds the one of mainland species by a factor of 9.5 and 8.1, respectively, for plants and vertebrates[5]. To date, efforts to assess scenarios of biodiversity changes in the context of future climate change rarely focus on island ecosystems[8,9]. Moreover, past island-based assessments of climate change impacts are limited to exposure to threat[10] and did not take into account the species characteristics enabling them or not to cope with climate change, which may have strong implications on the true vulnerability of species to climate change. This clearly limits our ability to prioritize conservation actions on islands.

Indeed, species' ability to cope with climate change impacts will depend on both internal and external factors such as their exposure to threat and the traits that allow them—or not—to cope with this exposure[10,11]. This ability (or lack thereof) to cope with the adverse effects of climate change can be summarized by their vulnerability to the threat. Since the Fourth Assessment of the Intergovernmental Panel on Climate Change (IPCC) in 2007 and earlier developments for natural hazards, vulnerability has been defined by three components—exposure, sensitivity, and adaptive capacity[12]—which have been adapted and applied to multiple biodiversity assessments[10,11,13–15]. Thus, even if the species' physical environment changes due to climate change (i.e. exposure), its climate change vulnerability is influenced by its ability to persist in situ (i.e. sensitivity) and adjust to the negative impacts of climate change (i.e. adaptive capacity). While identifying vulnerable species is essential, as it provides valuable information on those with a high likelihood of persistence and their location, this question remains largely ignored in the literature on islands ecology. Importantly, it can help develop effective conservation strategies to prevent possible species extirpation or extinction[15]. It can also allow us to identify the areas where species are less vulnerable to climate change and where climate refugia may be identified to protect species.

Since the adoption of the IPCC vulnerability definition, several approaches focussing on different vulnerability components have been developed to assess species vulnerability to climate change[8,9,15]. The studies that do investigate climate change impacts either used correlative approaches focussed only on the exposure component to estimate species' distributional changes under climate change[16–18] or combined them with mechanistic[19–21] and trait-based approaches[11,13], which are limited by data availability and often restricted to particular taxa or regions. Although the trait-based approach is currently the least frequently used in the literature to assess species vulnerability to climate change[8,9], species trait data are increasingly available for many taxa, which allows the application of such a method to insular species. However, many traits suffer from a choice of variable thresholds to establish vulnerability. For instance, for traits with continuous values, there is often no ecological

threshold to separate high and low scores[22]. To prevent the use of arbitrary thresholds, multicriteria decision analysis can be applied[23,24] for the quantitative measurement of vulnerability according to their relative positive and negative ideal solutions in a context of climate change.

By combining and adapting trait-based and quantitative vulnerability frameworks, we investigate here the vulnerability of 340 islands to climate change by 2050 based on 873 endemic mammal species (Fig. 1). Our analyses focus on mammals, because severe population declines following climate change have already been reported for this taxa[25]. Regarding the climate change vulnerability components, exposure is described by a local climate change metric while sensitivity and adaptive capacity are characterized by several species and insular characteristics that have been shown to be important response variables to climate change (Table 1). Specifically, we identify the hotspots of islands and archipelagos vulnerable to climate change. We then assess the influence of the component among exposure, sensitivity, and adaptive capacity on vulnerability to climate change to establish the drivers of island and archipelago vulnerability. Finally, we explore how species' ecological traits are associated with climate change vulnerability. Although our findings demonstrate that all islands and archipelagos show a certain level of vulnerability to future climate change, vulnerability hotspots are found in the Pacific Ocean. The spatial pattern of climate change vulnerability cannot be explained by exposure component alone, which is, therefore, not a reliable proxy to assess island vulnerability. This work reveals the importance of exploring climate change vulnerability at island scale in order to develop effective conservation strategies to prevent possible extinction of unique biodiversity.

## Results

**Island and archipelago vulnerability to future climate change.** We observed a spatial variation of vulnerability values across islands from 0.18 to 0.71 (Fig. 2a) and archipelagos from 0.25 (for Tasmania) to 0.67 (for New Hebrides) (Fig. 2b). Highly vulnerable islands (>0.5; 63% of all islands) were found in the Pacific Ocean, while the lowest vulnerability values were found for the islands of Japan, Tasmania, Sri Lanka, and Caribbean (Fig. 2a). When averaging the vulnerability values of islands for each archipelago, 6 of them (42%) were identified as highly vulnerable (>0.5): New Hebrides, Bismarck Archipelago, New Caledonia, Solomon Islands, Malay Archipelago, and Sulawesi (Fig. 2b). We found no correlation between climate change vulnerability and endemic species richness of islands and archipelagos (Supplementary Fig. 1). By decomposing the three vulnerability components, the level of exposure (mean ± s.d.: 0.25 ± 0.21) was lower than sensitivity (0.50 ± 0.20) and adaptive capacity (0.40 ± 0.14; Fig. 2c–e). We also found a significant positive relationship between sensitivity and exposure (Fig. 2c) but no significant relationship for the other two relationships at the archipelago scale (Fig. 2d–e). Eight archipelagos, including the six identified as highly vulnerable, were characterized by high sensitivity (>0.5) and low adaptive capacity (<0.5) values. Conversely, and strikingly, only two vulnerable archipelagos were identified as highly exposed (>0.5): Bismarck Archipelago and New Hebrides. Robustness analyses demonstrated that the spatial patterns of vulnerability and its components were robust to the alternative normalization methods and choice of representative concentration pathways (RCPs; Spearman's rho > 0.8; Supplementary Figs. 2 and 3). In addition, when one variable of a vulnerability component was removed from the analyses, its values were close to those observed with all variables, highlighting the robustness of our results. Nevertheless, the distribution patterns of vulnerability component values can differ when some of the variables are

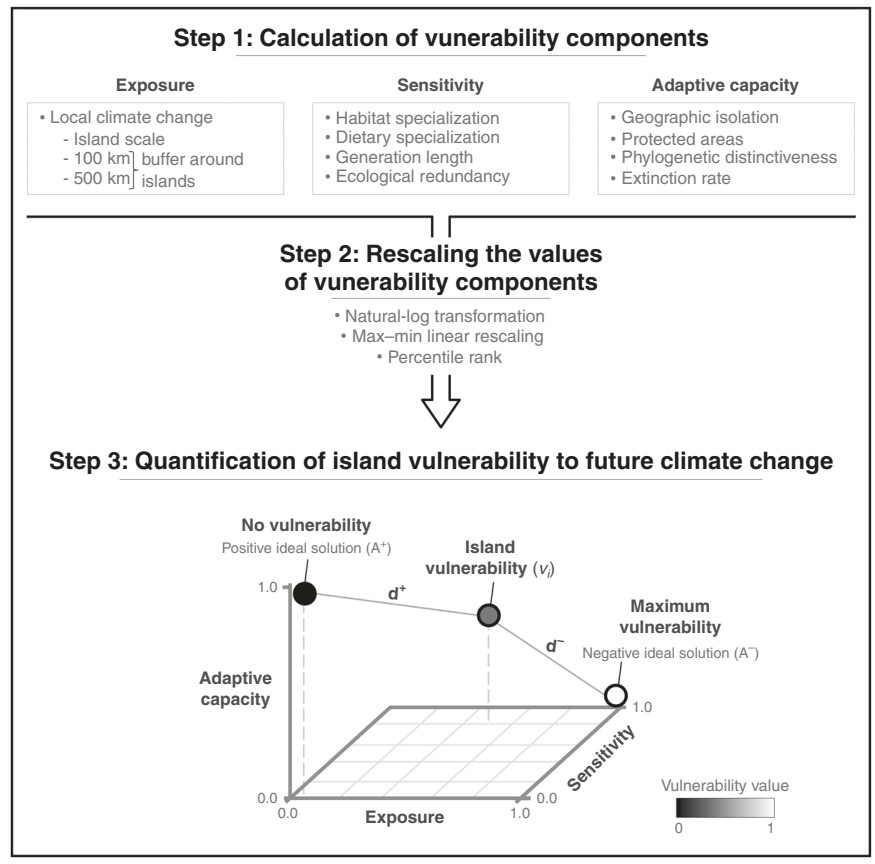

**Fig. 1 Overview of the step-by-step analytical framework.** See "Methods" section, Table 1, Supplementary Table 3, and Supplementary Fig. 11 for details.

**Table 1 Characteristics of species and islands that describe sensitivity and adaptive capacity of climate change.**

*Sensitivity*

| | |
|---|---|
| Habitat/dietary specialization | A more specialized species has a greater probability of negative responses in the face of climate change because of less likely to proliferate into new suitable climatic areas[39-42] |
| Generation length | Species with long generation lengths have a greater probability of negative responses in the face of climate change, as long generation times offer less potential for genetic adaptation (as well as lesser demographic response to conservation measures) compared to short generation times[7,21,39] |
| Ecological redundancy | Islands of low ecological redundancies will particularly suffer from high rates of co-extinctions and loss of key functions due to climate-change-driven disruptions of ecological interactions[7] |

*Adaptive capacity*

| | |
|---|---|
| Geographic isolation | Geographic isolation acts as a very efficient dispersal filter, drastically limiting the potential of species' responses to ecological shifts due to climate change by migration to other landmasses[7] |
| Protected areas | Promote community resilience to climate change and could provide suitable areas that promote colonization from and towards unprotected areas[37,65,66] |
| Phylogenetic distinctiveness | A more phylogenetically diverse species pool has a higher evolutionary potential to adapt and to persist in the face of climate change[69] |
| Extinction rate | Species that have evolved, and survived, in high disturbance environments should be more likely to persist in the face of new disturbances like climate change[34,71] |

removed. For example, sensitivity values tend to increase when dietary specialization or ecological redundancy are not considered, yet the archipelagos identified as highly or lowly sensitive to climate change remain the same (Supplementary Figs. 4 and 5).

**Association of climate change vulnerability and its components.** The spatial variation of vulnerability values among archipelagos can be explained by different factors emphasized in the principal component analysis (Fig. 3). The first principal component explains 33.5% of the variance and is strongly associated with increasing vulnerability (Pearson's $r = 0.98$) and exposure (Pearson's $r = 0.85$) and decreasing adaptive capacity (Pearson's $r = -0.73$). The second principal component explains 18.7% of the variability, with this axis mostly associated with sensitivity (Fig. 3; Pearson's $r = 0.71$; see Supplementary Fig. 6 and Supplementary Table 1 for more details). Vulnerability is mostly driven positively by exposure (Spearman's rho $= 0.85$) and negatively by adaptive capacity (Spearman's rho $= -0.71$) and, to a lesser extent, by sensitivity (Spearman's rho $= 0.41$). We can distinguish the different archipelagos and their vulnerability along the two axes (Fig. 3). For example, Solomon Islands displays some of the highest values for sensitivity compared to other archipelagos. We also observed that archipelagos with the more conservative values of climate vulnerability (i.e. Japan, Sri Lanka,

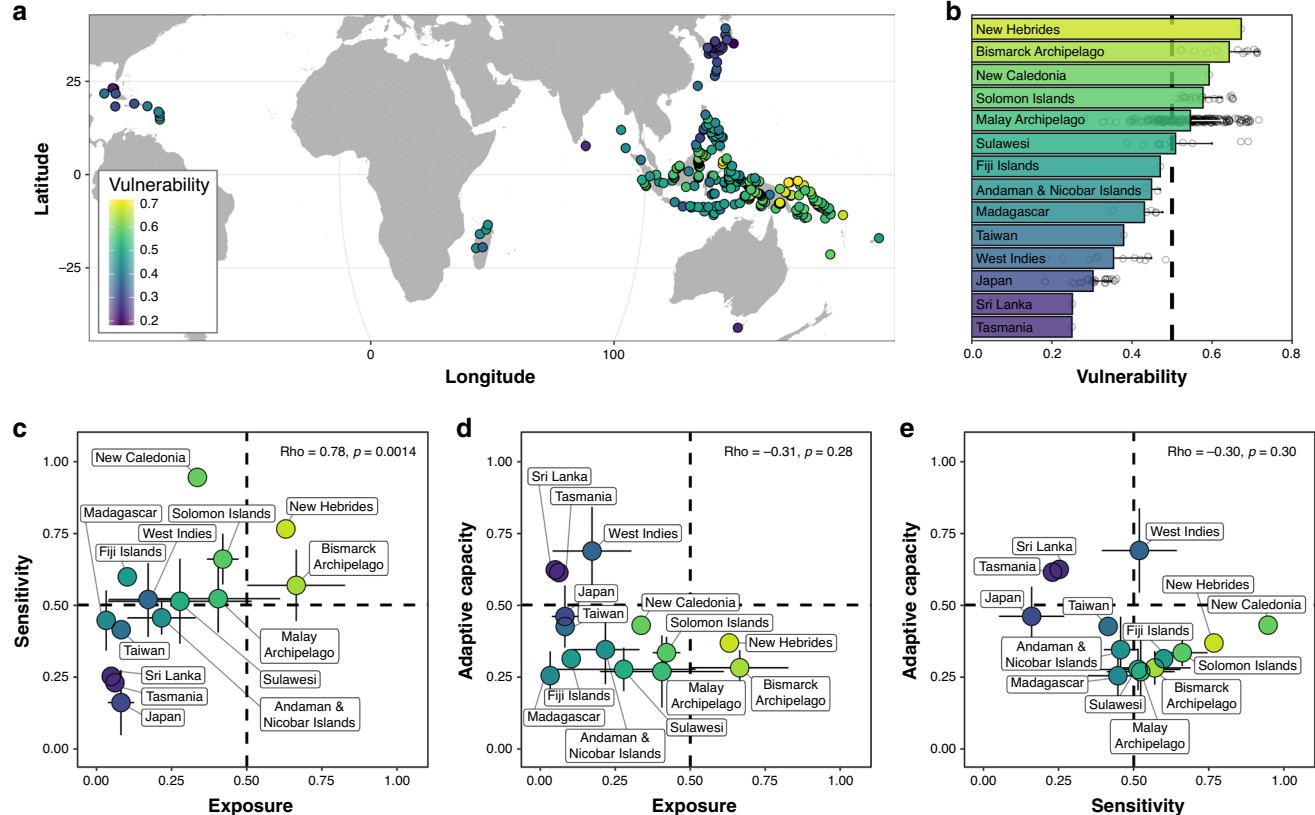

**Fig. 2 Vulnerability assessment and its components' relationship of endemic island mammals. a** Global gradient of island vulnerability to future climate change. **b** Vulnerability ranking of the 14 archipelagos. **c–e** Relationships between vulnerability components (exposure, adaptive capacity, and sensitivity) for the 14 archipelagos ($n = 340$ islands). In **b–e**, colour points or colour bars represent mean values and associated error bars represent standard deviation values. Spearman correlation coefficients are shown in **c–e**. A 3D visualization can be found at https://chart-studio.plotly.com/~BIOM/1/. Dashed lines on **b–e** represent an intermediate value of 0.5 of vulnerability or its components. Results are shown here for RCP 6.0—2050 and the basic standardization. The identification of islands and archipelagos is based on data from Weigelt et al.[55] (see Supplementary Table 2).

Tasmania, Madagascar, and West Indies), all have low values of exposure and high values of adaptive capacity, but the associated variables responsible for these differ (Fig. 3). For instance, in the West Indies, the adaptive capacity is associated with geographic isolation and extinction rate, while in Japan it is more associated with phylogenetic distinctiveness. The second principal component emphasizes that the vulnerability of West Indies and Solomon Islands is mostly driven by high sensitivity, while the opposite pattern is found for Japan.

**Species' ecological variables and climate change vulnerability.** Although sensitivity was an important factor for particular archipelagos (e.g. Solomon Islands and West Indies), it was generally not detected as the most important driver of vulnerability at the island scale. We thus tried to identify the ecological traits associated with high vulnerability at the species level. Among the four ecological traits of the sensitivity component, two of them were significantly positively correlated to vulnerability values: dietary specialization and generation length (Fig. 4a, b). This pattern can differ at the archipelago scale. For example, West Indies showed a negative relationship for dietary specialization (Spearman's rank correlation: $S = 244.11$, rho $= -0.11$, $p = 0.75$) and generation length ($S = 246.00$, rho $= -0.12$, $p = 0.73$) (Supplementary Fig. 7). Conversely, no relationship was found between vulnerability and ecological redundancy or habitat specialization at the global scale (Fig. 4c, d). However, we found different patterns depending on the archipelago considered. For West Indies, vulnerability was negatively associated with

ecological redundancy and habitat specialization, while the opposite was found for Japan (Supplementary Fig. 7).

**Discussion**
Our 2050 insular climate change vulnerability assessment revealed a clear pattern of spatial heterogeneity in terms of island and archipelago vulnerability. This pattern was not driven by species richness (Supplementary Fig. 1), while this is a common criterion to highlight the most important areas for biodiversity and to allocate limited conservation resources effectively[26]. Prioritization approaches based only on richness have already shown to mislead on which areas to protect, which suggests that it is important to include others factors such as vulnerability, both at current[27,28] and future time. Although insular vulnerability values were heterogeneous, all islands and archipelagos were predicted to be affected by future climate change. Therefore, climate change will be an additional threat for insular ecosystems that are already particularly threatened by current threats[29]. Surprisingly, we found that exposure, which is overall homogeneous within the islands (Supplementary Fig. 8), is not a key factor to explain future vulnerability to climate change. Indeed, only two archipelagos (Bismarck Archipelago and New Hebrides) were highly exposed to climate change, yet six archipelagos in the Pacific Ocean were potentially highlighted as highly vulnerable: Bismarck Archipelago, Malay Archipelago, New Caledonia, New Hebrides, Solomon Islands, and Sulawesi. This finding is particularly important as it highlights the potential high sensitivity of island faunas and their low adaptive capacity to explain

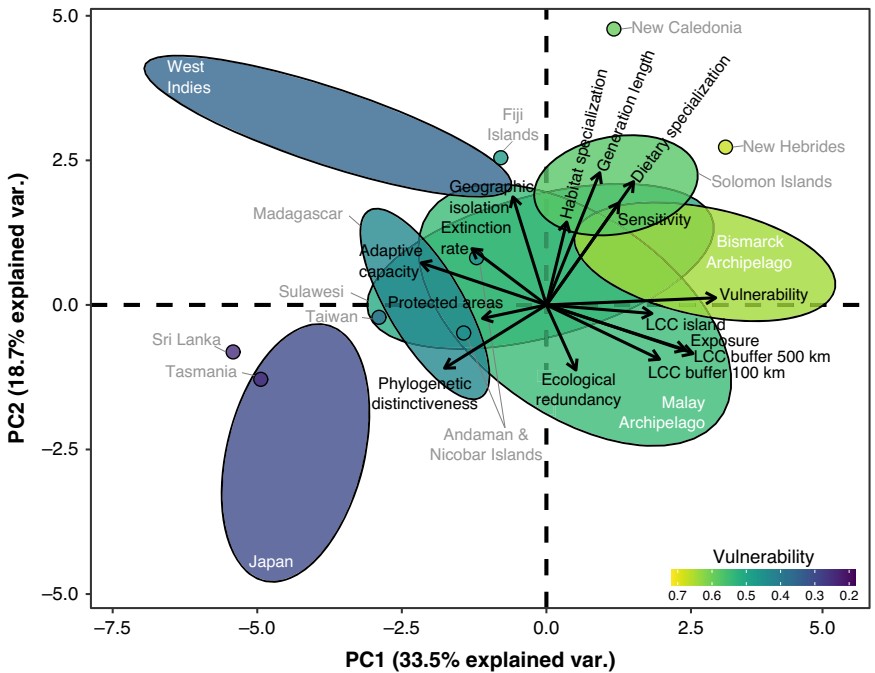

**Fig. 3 Descriptive pattern of vulnerability, its components (exposure, adaptive capacity, and sensitivity), and respective variables.** The principal component analysis biplot shows the factors explaining the spatial variation of vulnerability. Coloured ellipses: archipelagos with more than two islands; coloured points: islands of archipelagos representing one or two islands (see Supplementary Table 2). LCC refers to local climate change metric (see "Methods" section for further information). All variables and components were standardized by the basic standardization for RCP 6.0—2050. See Supplementary Fig. 6 for more information on PCA.

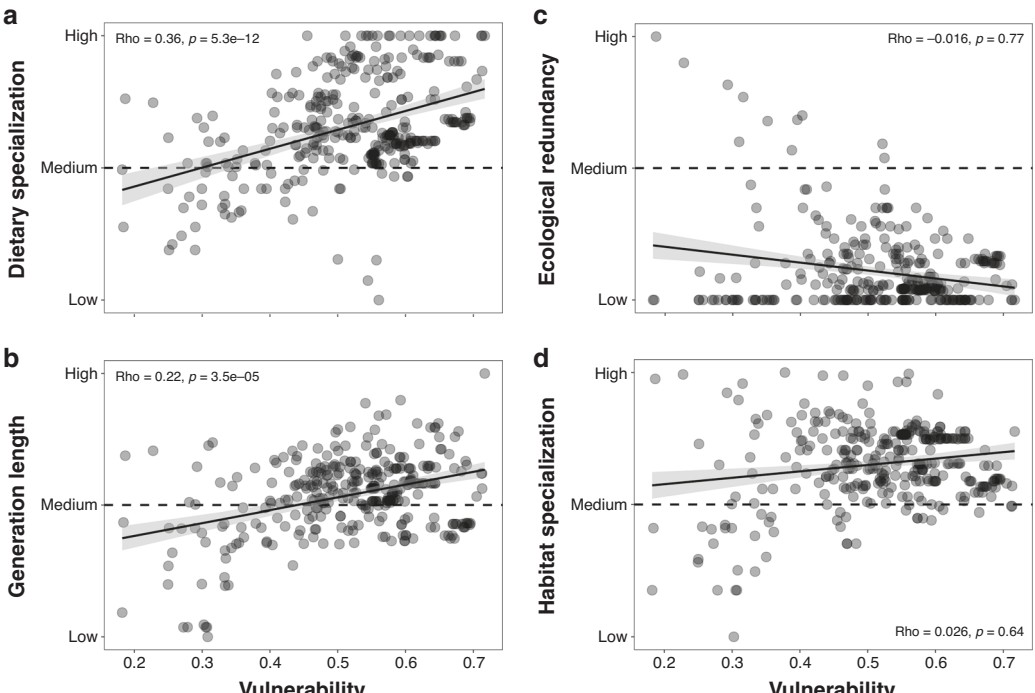

**Fig. 4 Relationships between ecological characteristics and vulnerability to climate change of insular endemic mammal pools.** Relationships between climate change vulnerability and **a** dietary specialization, **b** generation length, **c** ecological redundancy, and **d** habitat specialization (n = 340). Spearman correlation coefficients and smooth curves based on a linear method with error bands representing a 95% confidence level are shown. Results are shown here for RCP 6.0—2050 and the basic standardization.

vulnerability. On the one hand, Bismarck Archipelago, Malay Archipelago, and Solomon Islands include the islands of Papua New Guinea, which is already depicted as an important hotspot of climate change risk for mammals[30]. Future climate change could possibly disrupt and disaggregate extant species communities in this region[30]. On the other hand, to our knowledge, New Caledonia, New Hebrides, and Sulawesi have never been identified as vulnerable archipelagos to climate change based on mammals[30] or other taxonomic groups[11]. However, no climate change effect assessment has yet focussed on insular endemic species or the vulnerability of these islands. In order to prevent their extinctions, more attention should be paid in the future to this particular set of unique species showing to be at high vulnerability to climate change. Archipelagos characterized by both low values of exposure and high values of adaptive capacity such as West Indies and Tasmania, though restricted to endemic mammals, provide interesting insights that may allow us identify potential refugia of endemic mammals in the future.

Island exposure was positively associated with vulnerability to future climate change, but it was rarely the main driver. Our study thus confirms that the classical approach of focussing only on exposure can give a biased view of vulnerability to climate change by ignoring the biological characteristics of species that may significantly increase or reduce their vulnerability[9,11]. For example, it has been shown that worldwide mammals predicted to experience the greatest magnitudes of climate change across their existing geographic distributions have in many cases relatively broad tolerances to climate[31]. In our island context, we obtained a different pattern where exposure is correlated to the species' sensitivity to climate change, which can be partly explained by the focus on endemic mammals. Mammals are more likely to be habitat/diet specialists and so less able to tolerate even small changes of climate. Even so, the importance of historical factors (e.g. phylogenetic distinctiveness, extinction rate) or species' ecological characteristics seem to be more important than changing climate alone to define vulnerable islands/archipelagos. In addition, archipelagos may have similar climate change vulnerability or even similar levels for the three components, yet show important differences in terms of the variables driving them. A first example is West Indies and Japan, with similar values for adaptive capacity but with differences among variables constituting the adaptive capacity component. Compared to Japan, West Indies was characterized by a higher rate of species extinctions, which is mostly driven by biological invasions, agri-/aqua-culture, and overexploitation, not by climate change[29]. However, it has been showed that these drivers of extinction mostly targeted specialist species having high reproductive speed[32], just like climate change, which could render a part of the remaining community more resilient to climate change. Moreover, even if documented extinctions of vertebrates mostly date back to the 1500s, it was estimated that many species have succumbed within the past millennia due to human colonization[33] or possibly due to glacial–interglacial cycles[34]. By considering pre-1500 extinctions, the resilience capacity of insular species to climate change could be better assessed. Another example is Bismarck Archipelago and Sulawesi, both identified as vulnerability hotspots that have similar low capacity adaptive values but differ in terms of their protected area (PA) coverage. Bismarck Archipelago is characterized by a null PA coverage, while ~10% of Sulawesi area is under protection. Even though PAs may not meet the needs of species in the future, as was shown for the Azores islands[35], they could play a key role in the mitigation and/or adaptation to climate change[36]. Indeed, PAs promote less-disturbed ecosystems compared to ecosystems damaged by human disturbances and thus may promote community resilience to threats including climate change[37]. Overall, to truly assess the vulnerability of a system, all the components and their related variables need to be considered and explored.

Species' ecological characteristics may significantly influence their sensitivity and adaptive capacity as well as their vulnerability to face climate change. In our case, species with a strong dietary specialization and a long generation length were associated with high vulnerability values. Thus our analyses revealed that the species likely to first disappear from climate change in the most vulnerable islands are diet-specialized species with long generation lengths. Such species extinctions due to climate change could lead to a substantial trophic downgrading of island communities and imply severe consequences for ecosystem processes that depend on specific interactions[38]. These two variables have already been identified as response factors to climate change[39]. Indeed, species with a narrow diet have low ecological generalization and decreased food availability, meaning that they are less likely to proliferate into new suitable climatic areas[40–42]. By contrast, habitat specialization and the lack of ecological redundancy of the insular species pools were not especially associated with vulnerability to climate change, even though they were previously identified as such[23,39]. However, those studies focus on both continental and insular species; those traits might be identified because they are associated with continental species response to climate change (as the number of continental species is much higher than insular ones). We can question the similarity of responses to climate change between island and continental species. Generally, low ecological redundancy increases the vulnerability of a system to further change[43]. In our study, ecological redundancy was generally low among islands. Indeed, island ecosystems are often characterized by low species diversity, resulting in low ecological redundancy within communities[44]. Moreover, ecological redundancy was estimated using only endemic mammal species while disregarding other native species, which may artificially underestimate the true ecological redundancy for island communities. Accordingly, the low variability in terms of ecological redundancy in our study has not revealed a pattern associated with vulnerability. However, the global patterns differ at the archipelago scale. Investigating climate change vulnerability at a different scale will therefore contribute to better identifying the factors involved and thus guide conservation responses.

The choice of traits may partly influence species vulnerability and its components, as showed by robustness analyses, and could explain the wide range of outputs across the different trait-based assessments[8,22,45]. In our case, when some variables of sensitivity and adaptive capacity components (e.g. dietary specialization and ecological redundancy; extinction rate and PAs) are not considered, vulnerability component values tend to increase, yet the archipelagos identified as highly and lowly sensitive or characterized by low and high adaptive capacity to climate change remain the same (Supplementary Figs. 4 and 5). Previous studies showed that excluding intrinsic traits that determine species sensitivity and adaptive capacity can underestimate or overestimate vulnerability[46,47]. Further research is required to identify relevant species traits in an island context[22]. Interestingly, some of our results were congruent with previous studies by highlighting archipelagos that were already identified as hotspots of climate change risk for mammals, for example, Bismarck Archipelago, Malay Archipelago, and Solomon Islands, which include the islands of Papua New Guinea[11,30]. In addition, large-scale climate projections do not provide a fine enough resolution for the island scale, which does not account for geographical and climatic variations within high-elevated oceanic islands, for example[7]. Indeed, many oceanic islands have highly structured landscapes, which can generate numerous microclimatic conditions used as refugia for species and thus reduce their extinction

risk from climate change[48,49]. Thus we may have underestimated the opportunities for species to escape from climate change via the use of microclimatic conditions. However, large-scale climate projections are necessary for the global analysis of island vulnerability to climate change. There is a need to develop appropriate climate data and models at the island scale to examine and identify the microrefugia available for endemic species affected by climate change. Last, for a global view of island vulnerability to climate change, entire communities (i.e. not only endemic species or species from a specific taxonomic group) should be considered, as should other climatic components. Although temperature and precipitation changes are highly relevant[7], other climate change factors such as rising sea levels[50] or extreme climatic events[51] can highly impact insular biodiversity.

Our approach identified island and archipelago biodiversity vulnerable to future climate change. Specifically, we highlighted that appropriate conservation management strategies and actions must be implemented as a priority for Bismarck Archipelago, Malay Archipelago, New Caledonia, New Hebrides, Solomon Islands, and Sulawesi, all highly vulnerable to future climate change despite the fact that only two of them will be highly exposed to climate change by 2050. Because indirect impacts from human responses to climate change (e.g. altered agricultural activities, different fishing efforts, human migration, or targeted area protection) can influence the ability of species to cope, adjust, or disperse away from climate impacts, we call for further investigations to consider them in vulnerability assessments[52,53].

## Methods

**Global geographic distribution of insular endemic mammals**. We used occurrences of 873 mammal species, endemic to one (i.e. single-island endemics) or many islands (i.e. multi-island endemics) based on distribution information taken from the IUCN Red List[54]. By overlapping mammal occurrences with island data from Weigelt et al.[55], 340 islands from 14 archipelagos (Supplementary Table 2) with climatic information (see next section) were considered: each island harbours >5 species (mean ± s.d.: 15 ± 13 species; Supplementary Fig. 9).

**Calculation of vulnerability components**. To measure vulnerability, which is a function of three components including exposure, sensitivity, and adaptive capacity (Fig. 1), we proposed to adapt trait-based assessment frameworks[11,14] by combining them with a quantitative framework[23].

Exposure is based on a measure of local climate change for each pixel. Following Williams et al.[56], to quantify local climate change, we calculated the standardized Euclidean distances (SEDs) between current and future climates per grid point, as follows: $\text{SED}_{ij} = \sum_{k=1}^{6} \frac{(b_{ki} - a_{kj})^2}{s_{kj}^2}$, where $a_{kj}$ and $b_{ki}$ are the current and future means for climate variable $k$ at grid points $i$ and $j$ and $s_{kj}$ is the standard deviation of the interannual variability across the 30-year climate window. Following the methodology of Bellard et al.[57], we chose six different climate variables to calculate local climate change: annual mean temperature, maximum temperature of the warmest month, minimum temperature of the coldest month, annual precipitation, precipitation of the wettest month, and precipitation of the driest month. The standardization values were temperature seasonality for temperature variables and precipitation seasonality for precipitation variables. Standardizing each variable placed all climate variables on a common scale[58]. Current climate data were averaged from 1970 to 2000 at 30-s resolution (~1 km²) from the Worldclim database[59]. Future climate projections for 2041–2060 (hereafter 2050) were downloaded for five general circulation models from the Fifth Assessment Report of the IPCC[60]: CCSM4 (Community Climate System Model, version 4), HadGEM2-ES (Hadley Global Environment Model 2—Earth System), MIROC5 (Model for Interdisciplinary Research on Climate, version 5), MRI-CGCM3 (Meteorological Research Institute Coupled Global Climate Model, version 3), and NorESM1-M (Norwegian Earth System Model 1—medium resolution), using two RCPs: RCP 6.0 (assumed global average increase of 2.85 ± 0.62 °C in mean annual temperatures) and RCP 8.5 (assumed global average increase of 4.02 ± 0.80 °C). All islands considered in the study are represented by at least 10 grid points of climate data (mean ± s.d.: 14,517 ± 85,973). Using this metric, we first calculated the SED values between the current and future climate data for each grid point of a given island and then computed the average value for each island (mean ± s.d./min–max: 11.33 ± 8.35/0.86–69.78 [RCP 6.0]; 13.89 ± 9.74/0.75–67.74 [RCP 8.5], based on all islands). The higher the value, the higher the climate dissimilarities between two periods, indicating a high exposure to climate change in the future[56,61]. Because a species in a given island may find suitable climatic conditions in nearby islands, we also calculated the SED values within a 100- and 500-km

buffer around each island (corresponding to the apparent long-distance dispersal limitation of large and small mammals[44]), and as before, the average value was obtained at the island scale.

For a given island, we characterized the sensitivity of climate change of all its inhabiting endemic mammals based on four species attributes: habitat specialization, dietary specialization, generation length, and ecological redundancy (Table 1 and Supplementary Table 3), which we calculated as an average value for each island. First, habitat and dietary specializations were defined as the number of habitats and diets for each endemic mammal species. Habitat information was extracted from the IUCN Habitat Classification Scheme[54] and dietary data from the EltonTraits database[62]. Habitat and dietary specialization among mammals has been shown to be an important response factor to climate change, since a more specialized species has a greater probability of negative responses[39]. Then generation length is defined as the average age of parents of the current cohort, reflecting the turnover rate of breeding individuals in a population. Such data were extracted from the database of Pacifici et al.[63]. Long generation lengths have been shown to be associated with a heightened extinction risk under climate change[21], especially among mammals, in which low reproductive rates (linked to long generation lengths) showed a greater probability of negative responses to climate change[7,39]. Last, ecological redundancy was calculated as the number of species that shared similar combinations of ecological trait values[64]: here, main diet, foraging niche, foraging period, habitat niche breadth, and body mass. Habitat information was extracted from the IUCN Habitat Classification Scheme[54] and the other four variables data from the EltonTraits database[62]. All trait values were transformed into categorical nature to identify ecological trait combinations and thus grouped species sharing the same trait values into these entities[64]. It has been shown that islands characterized by low redundancy values (i.e. different ecological profiles supported by unique species) will suffer from high rates of co-extinctions and loss of key functions, due to climate-change-driven disruptions of ecological interactions[7].

To characterize the adaptive capacity of mammals on each island, four attributes were considered: geographic isolation of the island, percentage of PAs, phylogenetic distinctiveness of the endemic mammal species, and extinction rate on the island (Table 1 and Supplementary Table 3). First, geographic isolation was based on the proportion of surrounding land mass, which is the sum of the proportion of land mass within buffer distances of 100, 1000, and 10,000 km around the island perimeter[55]. Geographic isolation may also influence species adjustment to climate change, because an isolated island has fewer nearby potential refuges where the species might find suitable conditions and thus escape from climate change[7]. Geographic isolation was negatively correlated to other island environmental variables such as elevation and area, factors also linked to potential habitat availability for species to escape from climate change[7]. Then the ratio of island area covered by PAs was also used to characterize the adaptive capacity of islands. PA is also an indication of the preservation of the habitats and ecosystems to perturbations, which contributes to mitigate the vulnerability to any threat and also ensure habitats' continuity and thus facilitate species range shifts in the face of climate change. Indeed, protected ecosystems promote better community resilience to climate change than ecosystems damaged by human disturbances[37]. Moreover, PAs could provide suitable areas that promotes colonization from and towards unprotected areas[65,66]. To estimate protection status, PA spatial information was collected from the World Database on Protected Areas (available at http://protectedplanet.net/; see Supplementary Methods for more details). Furthermore, phylogenetic distinctiveness was defined as the average evolutionary isolation of endemic mammal species within each island, which is based on the fair proportion index that quantifies the number of relatives of each species and their phylogenetic distance[67]. We used an updated phylogeny for mammals[68] that was restricted to the species considered in this study. We analysed species phylogenetic distinctiveness based on the 1000 trees available[68] with randomly resolved polytomies to account for phylogenetic uncertainty. Such an index can be informative about the evolutionary potential of species pools, and so adaptive capacity in the context of climate change[69]. Indeed, more phylogenetically diverse species pools may have a higher evolutionary potential to adapt environmental perturbations such as climatic change[70]. Last, the extinction rate represents the ratio between the number of vertebrate extinctions since the year 1500 CE and the total species richness of vertebrates for each island[54]. The rate of species extinctions per island represents the potential resistance of the species to perturbations. Species that are more prone to extinctions because of intrinsic factors are more likely to disappear rapidly after the initial exposure to perturbations, like climate change, while extinction-resistant species are more likely to persist[34,71]. If the extinction rate was already high for a given island, we followed the filter hypothesis[71], according to which species that are highly sensitive to extinction have already disappeared, while the remaining species are more likely to resist to future climate change.

For each vulnerability component (exposure, sensitivity, and adaptive capacity), we tested for correlations between variables. No correlation was found (Spearman's $r < |0.7|$; Supplementary Fig. 10). We conducted robustness analyses to test the effects of selected variables. Specifically, we ran congruence analyses between distribution data of vulnerability components using all variables and using all combinations with one variable removed to test whether a given variable drove the observed patterns of exposure, sensitivity, and adaptability patterns at island scale. We also investigated the robustness of vulnerability ranking among archipelagos.

**Standardization of island vulnerability to climate change.** We standardized the values of each individual variable of exposure, sensitivity, and adaptive capacity to a 0–1 range to create unit-less metrics (Fig. 1). We tested three types of transformations that are usually used for the 0–1 standardization[72–74]. First, we used a basic standardization, a max–min linear rescaling $\left(\frac{x_i - x_{min}}{x_{max} - x_{min}}\right)$. Second, we used a natural-log transformation (In $[x + 1]$) before max–min rescaling, thereby compressing the upper end and expanding the lower end of the range. This process minimizes the influence of extremely high values that could be spurious but influential. Third, we used a cumulative distribution function to replace each value with a score reflecting its percentile rank relative to all others. For any given level of value, the cumulative distribution function approach resulted in a uniform number of values, whereas the linear and log transformations resulted in a heterogeneous number of values. After standardizing each variable, we calculated the sum of the variables of each component (i.e. exposure, sensitivity, and adaptive capacity) and re-standardized the obtained values of exposure, sensitivity, and adaptive capacity.

**Quantification of island vulnerability to climate change.** Finally, based on the method of Parravicini et al.[23], we calculated a measure of vulnerability for each island. A multicriteria decision analysis was applied, specially the TOPSIS method (Technique for Order Preference by Similarity to an Ideal Solution). This technique ranks alternatives according to their relative distance to positive and negative ideal solutions, which represent the conditions obtained when the criteria have extreme values[75]. In our case, vulnerability was considered as a function of three criteria: exposure and sensitivity, both expected to favour vulnerability, and adaptive capacity, expected to reduce vulnerability (Fig. 1 and Supplementary Fig. 11). Hence, the positive ideal solution (i.e. no vulnerability) corresponds to the condition in which exposure and sensitivity to climate change are minimized and adaptive capacity is maximized. Nevertheless, the negative ideal solution (i.e. maximum vulnerability) corresponds to a condition of minimum adaptive capacity and maximum exposure and sensitivity to climate change. Vulnerability for each island was thus expressed as the relative distance to the positive and negative ideal solutions, ranging from 0 to 1 to reflect low to high vulnerability (Supplementary Fig. 11).

All analyses were performed using the R software (version 3.6.0)[76].

**Reporting summary.** Further information on research design is available in the Nature Research Reporting Summary linked to this article.

## Data availability

All data that support the analyses and findings of this study are public and listed hereunder. Current and future global climate data are available at http://www.worldclim.com/version2. Data about island geographic isolation and spatial distribution are available at https://datadryad.org/stash/dataset/doi:10.5061/dryad.fv94v. Protected area data are available at http://protectedplanet.net/. Species phylogeny is available at https://megapast2future.github.io/PHYLACINE_1.2/. Species generation lengths are available at https://datadryad.org/stash/dataset/doi:10.5061/dryad.gd0m3. Data about species habitat, extinction status, and spatial distribution are available at https://www.iucnredlist.org/. Species diet data are available at http://www.esapubs.org/archive/ecol/E095/178/metadata.php. The two last data sources were also used to compute ecological redundancy. Source data are provided with this paper.

## Code availability

All code necessary to conduct climate change vulnerability assessment is available for download at https://github.com/CamilleLeclerc/Vulnerability.

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

## Acknowledgements

We are thankful to Boris Leroy for his valuable comments on the methodology, as well as to Victoria Grace for her corrections of the English. This work was supported by grants from the ANR (14-CE02-0021-01), the AXA Research Fund Chair for Invasion Biology, and the Biodiversa Eranet AlienScenarios. More generally, this work has been funded by our salaries as French agents and by the CNRS funding for young researcher.

## Author contributions

C.L. and C.B. conceived the idea for this manuscript with input from F.C. C.L. collected the data and performed the analysis. C.L. wrote the first draft, and all authors contributed to writing the manuscript.

## Competing interests

The authors declare no competing interests.
