## [Peer Review File · Nature Communications]

Reviewers' comments:

Reviewer #1 (Remarks to the Author):

This manuscript examines the vulnerability of selected island ecosystems to future climate change through a rigorous and multi-faceted analysis of climate change exposure and the sensitivity and adaptive capacity of native mammal species to potential changes. The data used are rather limited in scope, with only islands from 14 archipelagos, and examined only one major taxon: mammals. However, given these limitations, I do believe that this paper presents a new and promising approach to understand the ways that climate change may play out to effect different insular communities differently. I believe this paper will be of interest to the conservation and climate change biology communities.

I have three points of critique I will list below:

1. It is my impression that most ecologists downplay the potential impacts of climate change on tropical islands, aside from sea-level rise. I am interested in further exploring the findings stated on line 152: "Island exposure was positively associated with vulnerability to future climate change, but it was rarely the main driver. Indeed, among the six archipelagos identified as highly vulnerable, only two showed a high exposure to future climate change..."

I think this is a very interesting point, because it elevates the characteristics of the island faunas themselves, and their potential high sensitivity and low adaptive capacity in the face of this change. This is the most important point of the paper, as it moves our thinking into a multi-faceted look at climate change impacts.

2. I would like to know how well does the ranking of vulnerability across islands compare to the raw biodiversity? I.e. do biodiversity hotspots capture any element of the vulnerability to future change? Given that much conservation planning is still based on raw biodiversity counts, how does this new vulnerability information help to prioritize conservation efforts.

3. I know that many of these islands have relatively few species of mammals, since mammal distribution on islands is strongly limited by dispersal. I am curious what this analytic approach would show for all vertebrates on the islands. At present, with only 5+ species of mammal in each island 'community', I think the generalizability of this study is relatively weak.

Methods section comments:

1. What is the effect of averaging the climate change data to the island scale? Would it be better to use the gridded data to represent the island's heterogeneous environment?

2. The prior extinction rate is an important factor to consider. It looks like the IUCN red list data were used to measure past extinctions. However, these data only record known extinct species since the year

1500 CE. I would recommend that the authors at least acknowledge the potential importance of pre-1500 extinctions in establishing a filter effect, as 1) many of the islands have been inhabited by humans and their commensals for millennia, and 2) where data are available, a high rate of extinction before 1500 has been recorded among island vertebrates.

Other comments:

1. Is the data publicly shared? – yes all data appears to be from publicly-accessible sources.
2. The figures are very nicely done.

Reviewer #2 (Remarks to the Author):

Dear Editor,

I reviewed the ms entitled “Future climate change vulnerability of islands” by Leclerc et al.

The topic is interesting, the ms is well written and provide important information. The analysis is well conducted and I recommend publication of this ms.

The only aspect that this ms is lacking, in my opinion, is a sensitivity analysis or a measure of how stable the results are. Vulnerability analysis is, by definition, quite subjective because the quantitative assessment of exposure, sensitivity and adaptive capacity is subjective by definition.

The authors made some reasonable choices to define these variables. However, the impact of such subjectivity on their results is not assessed. I provide some examples below.

Sensitivity

While habitat specialization, dietary specialization and generation length are species-level parameters that may depict the tendency of a species to local extinction, the authors also account for ecological redundancy. This variable is not related to the species, but it is more related to a functional group. Is a species more vulnerable if it belongs to a non-redundant functional group? This choice may make sense to emphasize the risk of losing one unique combination of functional traits at community level, but it is quite subjective and likely mix to different levels of sensitivity, i.e. species level and community level. What are the patters of sensitivity if this variable is removed?

Adaptive capacity

Here the authors combine species level estimates with island-level estimates, such as phylogenetic distinctiveness and vertebrate extinction rate for the island or their geographic isolation. Again, these choices may make sense, but are quite subjective. An evaluation of the robustness of the results will help to understand which patterns are more prone or less prone to such subjectivity.

That said I believe this is a very nice study

Reviewer #3 (Remarks to the Author):

Dear colleagues,

Your study on the vulnerability of islands to climate change (CC) is quite interesting, well presented, and certainly timely. Your definition of “vulnerability hotspots” is highly informative and could indeed be useful to establish conservation actions and priorities. As such, your study can be certainly interesting for the broad readership of Nature Communications. That said, I do have a few interrelated concerns on your approach that can hinder the potential importance of your study by compromising your conclusions.

I was particularly concerned by the provided theoretical basis of your vulnerability metric. First, there is no clear/explicit mention on why “overall vulnerability...can have strong implications on results” (Lines 38-39). Which led me to my main concern about it: “overall vulnerability” is not the same as “climate change vulnerability”. While the former may be related to any threat, the latter is specific to CC. Accordingly, the components making up the integrated measure of vulnerability should indeed be directly related to the “...ability (or lack thereof) [of species] to cope with the adverse effects of climate change...”. However, there is not a clear/explicit justification on how do the chosen characteristics/traits inform us about the actual “sensitivity” and “adaptive capacity” of islands/species to CC. You do provide some of these “justifications” on your methods section but I believe these should come clearly in the introduction. More importantly, given that “adaptive capacity” and “exposure” were the most important components defining islands/archipelagos vulnerability, and that for most “vulnerable archipelagos” the exposure component was not that important, the need to provide an explicit (empirical or at least theoretical, with possibly the latter being more attainable) link between “adaptive capacity”, the specific traits considered and their relationship with climate change is fundamental.

Otherwise, it could be argued that all of these traits could have been measured and described to define islands/species vulnerability to any threat without having to rely on CC. That is, you would have found “a clear pattern of spatial heterogeneity in terms of island and archipelago vulnerability” without even considering exposure to CC simply as a result of islands/species intrinsic characteristics. Moreover, some of these traits are perhaps more relevant to other threats rather than to CC. For example, “protection status” may say nothing on CC adaptability as has been repeatedly suggested (e.g. Hannah 2008; Monzon et al. 2011; Melillo et al. 2016). Similarly, extinction rate (particularly in the way you calculated it, see below) and phylogenetic distinctiveness may be silent regarding CC adaptability. Please, don't get me wrong, I'm not saying the these traits (and the other ones you used) may not be related in some way to CC adaptability or that your approach is not valid, but only that the explicit justification for doing so is lacking from your manuscript.

Related to the importance of “adaptive capacity” vs exposure, taken at face value, could it then be said

that “historical factors” (phylogenetic distinctiveness, extinction rate) may be more important than changing climate alone (exposure: changes in the species’ physical environment) to define vulnerable islands/archipelagos? In any case, such possibility could be considered in your discussion section.

A somewhat less important issue is your “temporal window”. You modeled climate change by 2050, which could be relatively short considering the scale of species responses and threat classification. For instance, to be considered extinct, species must have not been seen by >50 years. In addition, perhaps the adverse effects of CC in the next 30 years may not be observable and thus your “predictions” may be overstated. Again, I’m not saying they are, but that you could justify your choice of time period more convincingly. From what I remember, in the “Global Change” literature and particularly in the “Ecological Niche/Species Distribution Modeling” (ENM/SDM) literature, climate projections consider two or three time windows (e.g. 2050, 2070, 2100) to evaluate potential impacts at different time periods while considering species’ responses at broader time scales.

Minor issues

Introduction

I believe that contextualizing your setting against known patterns in the continents/mainlands would be much informative for the reader. For instance, how “particularly prone” are island species in comparison to continental/mainland species?

Lines 34-35. 20% refers to island endemics or all species or all endemic species worldwide?

Lines 60-62. To support current validity of this argument, which is certainly true, I think you could cite more current literature. Otherwise, it gives the impression that this was done in the past and we are not sure what is being done nowadays.

Line 76. “...high vulnerability of this taxa” compare to what?

Results

Fig. 2b. You could include a vertical line at 0.5 in the vulnerability axis, just for reference.

Figs. 2c,d. I wonder if “Exposure” would be better placed at the X-axis, considering that this is an external measure that could be independent of the species whereas the other ones (sensitivity, adaptive capacity) include intrinsic properties of the species. Just an idea.

Lines 105-106. Yes, but by how much? Sure, the details are in SuppTable 3, but could you provide some description on the magnitude of the components’ effects?

Lines 112-113. All of these characteristics are independent of climate, right?

Discussion

Lines 136-141. But, only two of these were highly “exposed”.

Line 147. “...high risk of extinction following climate change...”, but other factors/traits (intrinsic and perhaps unrelated to climate or indirectly so) determined their “vulnerability” and not only “exposure” per se.

Lines 155-158. Agree, but this would be valid only if clearly justify the link between the traits used and

climate change “resistance”. See my main concern above.

Line 163. “Species extinction in this region...” which region? West Indies or Japan?

Lines 177-180. So, why use it in the first place? Again, the need to provide a stronger justification is fundamental.

Lines 180-181. Thus, a stronger justification on your choices is needed!

Lines 195-196. “...which questions the similarity of responses...for island and continental species...” why? Not clear, please explain.

Lines 213-214. “...highlighting archipelagos that were already identified...” For example? Please, repeat them here just for clarity.

Line 227. Just wondering, and based on your previous work, why not include “sea level rise” in your exposure component?

Lines 231-232. Again, only two of these are actually “exposed” to CC.

Methods

Exposure: how much of this “exposure” would species actually suffer? Any idea? Is there any data/result (e.g. from ENMs/SDMs) that you could use to contextualized (relative to...) the magnitude of “climate change” (based on your measured distance) to the known/expected tolerance of mammal species? Any discussion on this issue, if possible, would be very informative.

Lines 276-278. In the same vein, how high is “high”? Any reference value for comparison?

Lines 289-296. Some of this could be in the intro (justification).

Adaptive capacity: Why this phylogeny? Any justification based on your aims? What about using Bininda-Emonds et al. (2007 *Nature*; Fritz et al. 2009 *EcolLett*) phylogeny, Hedges et al. (2015 *MolBiolEvol*) or even Upham et al. (2019 *PLoS Biol*)? This is particularly relevant given that your metric depends on phylogenetic distances/branch lengths that were particularly highlighted by the original authors (Faurby & Svenning) as problematic: “...analyses using the resulting phylogeny should focus on the topology rather than on branch lengths” (pg. 16 in Faurby & Svenning 2015).

Line 323. What is the meaning of “resistance capacity”? Please, explain.

Lines 323-328. Extinction rate: how representative it is regarding your dataset? Why not considered only mammals instead of vertebrates?

Lines 328-331. How? Please, explain. Did you obtain a species-level (then averaged across species) or an island-level value?

References:

Monzón, J., Moyer-Horner, L., & Palamar, M. B. (2011). Climate change and species range dynamics in protected areas. *Bioscience*, 61(10), 752-761.

Hannah, L. (2008). Protected areas and climate change. *Annals of the New York Academy of Sciences*, 1134(1), 201-212.

Melillo, J. M., Lu, X., Kicklighter, D. W., Reilly, J. M., Cai, Y., & Sokolov, A. P. (2016). Protected areas' role in climate-change mitigation. *Ambio*, 45(2), 133-145.

COMMENTS FROM REVIEWER #1

This manuscript examines the vulnerability of selected island ecosystems to future climate change through a rigorous and multi-faceted analysis of climate change exposure and the sensitivity and adaptive capacity of native mammal species to potential changes. The data used are rather limited in scope, with only islands from 14 archipelagos, and examined only one major taxon: mammals. However, given these limitations, I do believe that this paper presents a new and promising approach to understand the ways that climate change may play out to effect different insular communities differently. I believe this paper will be of interest to the conservation and climate change biology communities.

→ We thank the reviewer for her/his positive appreciation of the study.

I have three points of critique I will list below:

1. It is my impression that most ecologists downplay the potential impacts of climate change on tropical islands, aside from sea-level rise. I am interested in further exploring the findings stated on line 152: “Island exposure was positively associated with vulnerability to future climate change, but it was rarely the main driver. Indeed, among the six archipelagos identified as highly vulnerable, only two showed a high exposure to future climate change...” I think this is a very interesting point, because it elevates the characteristics of the island faunas themselves, and their potential high sensitivity and low adaptive capacity in the face of this change. This is the most important point of the paper, as it moves our thinking into a multi-faceted look at climate change impacts.

→ Again, we thank the reviewer for this positive appreciation on one of our main. We have now further highlighted the importance of this result (abstract L22-23 and discussion e.g. L162-169 and 181-184).

2. I would like to know how well does the ranking of vulnerability across islands compare to the raw biodiversity? I.e. do biodiversity hotspots capture any element of the vulnerability to future change? Given that much conservation planning is still based on raw biodiversity counts, how does this new vulnerability information help to prioritize conservation efforts.

→ We now analyzed the relationship between climate change vulnerability and species richness (see Supplementary Fig. 1). We did not find any correlation between climate change vulnerability and endemic species richness of islands and archipelagos, and mentioned it into Results section (L100-102). We discussed this result in connection with conservation efforts and prioritization in the discussion (L154-159).

3. I know that many of these islands have relatively few species of mammals, since mammal distribution on islands is strongly limited by dispersal. I am curious what this analytic approach would show for all vertebrates on the islands. At present, with only 5+ species of mammal in each island ‘community’, I think the generalizability of this study is relatively weak.

→ We agree that it would be interesting to expand this study to other vertebrates such as birds, reptiles or amphibians. While exposure values and many components of adaptive capacity will be similar among terrestrial vertebrates, most sensitivity values require a tremendous amount of data that are currently not available for all vertebrates; this is particularly true for reptiles and amphibians. Regarding birds, more data are available; however, because of the mobile behavior of many species (seasonality), it is more difficult to establish their exposure depending on their behavior and thus their overall vulnerability. That is why we decided to focus our study on endemic mammals. It is also noteworthy that the limited dispersal abilities mentioned by Reviewer #1 makes mammals an especially

interesting group to study in terms of vulnerability. That being said, we hope that Reviewer #1 will agree that our results for mammals call for particular attention for all vertebrates regarding their vulnerability to climate change, and though not directly applicable to all other groups, they nonetheless underline particularly well the need to consider all the components of climate change vulnerability.

Methods section comments:

What is the effect of averaging the climate change data to the island scale? Would it be better to use the gridded data to represent the island's heterogeneous environment?

→ This is indeed an interesting and valid point and we understand the concern of reviewer #1. While using gridded data instead of island average is possible for climate, this is not the case for other variables such as extinction rate or sensitivity components that are only available at the island or species scale. In this context, we preferred to assess the climate change heterogeneity per island as it allows to keep all the variables at the same scale (see Supplementary Fig. 8). So on, we used the coefficient of variation index which has been used in previous studies to assess environmental heterogeneity (Stein & Kref (2015)). We found that the distribution of climate change heterogeneity values does not allow the islands to be discriminated, and the majority are characterized by a low climate change heterogeneity. In addition, climate change heterogeneity is – expectedly – correlated with island area (Spearman's rank correlation: $S = 1973294$, $\rho = 0.70$, $p < 2.2e-16$), elevation ($S = 1512262$, $\rho = 0.77$, $p < 2.2e-16$) and geographic isolation ($S = 10084645$, $\rho = -0.54$, $p < 2.2e-16$). These three insular environment variables are also correlated and one of them (i.e. geographic isolation) was already considered in our analyses. Because we want to ensure non-collinearity between our variables, we did not include it directly in our analyses but as a supplementary analysis. We thus expect that island's heterogeneous environment have been captured by geographic isolation. Thus, we mentioned about the climate change homogeneity within islands L162-164.

References:

Stein, A., & Kref, H. (2015), Terminology and quantification of environmental heterogeneity in species-richness research. *Biological Reviews* 90, 815-836.

2. The prior extinction rate is an important factor to consider. It looks like the IUCN red list data were used to measure past extinctions. However, these data only record known extinct species since the year 1500 CE. I would recommend that the authors at least acknowledge the potential importance of pre-1500 extinctions in establishing a filter effect, as 1) many of the islands have been inhabited by humans and their commensals for millennia, and 2) where data are available, a high rate of extinction before 1500 has been recorded among island vertebrates.

→ We agree that extinctions pre-1500 are also important to consider in establishing a filter effect, which is now acknowledged within the discussion (L202-205).

Other comments:

1. Is the data publicly shared? – yes all data appears to be from publicly-accessible sources.
2. The figures are very nicely done.

→ Thank you for your positive comment about the figures. We confirm that the data are publicly available.

COMMENTS FROM REVIEWER #2

Dear Editor,

I reviewed the ms entitled “Future climate change vulnerability of islands” by Leclerc et al. The topic is interesting, the ms is well written and provide important information. The analysis is well conducted and I recommend publication of this ms. The only aspect that this ms is lacking, in my opinion, is a sensitivity analysis or a measure of how stable the results are. Vulnerability analysis is, by definition, quite subjective because the quantitative assessment of exposure, sensitivity and adaptive capacity is subjective by definition. The authors made some reasonable choices to define these variables. However, the impact of such subjectivity on their results is not assessed. I provide some examples below.

Sensitivity: While habitat specialization, dietary specialization and generation length are species-level parameters that may depict the tendency of a species to local extinction, the authors also account for ecological redundancy. This variable is not related to the species, but it is more related to a functional group. Is a species more vulnerable if it belongs to a non-redundant functional group? This choice may make sense to emphasize the risk of losing one unique combination of functional traits at community level, but it is quite subjective and likely mix to different levels of sensitivity, i.e. species level and community level. What are the patters of sensitivity if this variable is removed?

Adaptive capacity: Here the authors combine species level estimates with island-level estimates, such as phylogenetic distinctiveness and vertebrate extinction rate for the island or their geographic isolation. Again, these choices may make sense, but are quite subjective. An evaluation of the robustness of the results will help to understand which patterns are more prone or less prone to such subjectivity.

→ We agree with this suggestion and have now added sensitivity analyses of vulnerability components (see Supplementary Fig. 4 and 5). More particularly, when one variable of a vulnerability component was removed from the analyses, its values were close to those observed with all variables, highlighting the robustness of our results. Nevertheless, the distribution patterns of vulnerability component values can differ when some of the variables are removed. For example, sensitivity values tend to increase when dietary specialization (species level) or ecological redundancy (community level) are not considered, yet the archipelagos identified as highly or lowly sensitive to climate change remain the same. Similarly, adaptive capacity values tend to increase when extinction rate (community level) or protected areas (island level) were not considered, but this has no impact on the archipelagos identified as having a high or low adaptive capacity. We mentioned it in Methods and Results part (lines 111-117, 384-389) and discussed these results lines 242-251.

That said I believe this is a very nice study

→ We thank the reviewer for her/his positive appreciation of the study.

COMMENTS FROM REVIEWER #3

Dear colleagues,

Your study on the vulnerability of islands to climate change (CC) is quite interesting, well presented, and certainly timely. Your definition of “vulnerability hotspots” is highly informative and could indeed be useful to establish conservation actions and priorities. As such, your study can be certainly interesting for the broad readership of Nature Communications. That said, I do have a few interrelated concerns on your approach that can hinder the potential importance of your study by compromising your conclusions.

→ We thank the reviewer for her/his positive appreciation of the study and have responded to the concerns below, hopefully satisfactorily.

I was particularly concerned by the provided theoretical basis of your vulnerability metric. First, there is no clear/explicit mention on why “overall vulnerability...can have strong implications on results” (Lines 38-39).

→ This is indeed an important point, as it is key to our main message. We rewrote this sentence to explain why focusing only on climate change exposure may lead to erroneous results and conclusions if we fail to take into account climate change vulnerability (L41-47).

Which led me to my main concern about it: “overall vulnerability” is not the same as “climate change vulnerability”. While the former may be related to any threat, the latter is specific to CC. Accordingly, the components making up the integrated measure of vulnerability should indeed be directly related to the “...ability (or lack thereof) [of species] to cope with the adverse effects of climate change...”.

→ We agree with the reviewer both that it is important and that overall vulnerability would have meant “including all threats”, while we did restrict vulnerability to CC threat. To avoid confusion, we replaced “overall vulnerability” by “climate change vulnerability” throughout our manuscript. We thank the reviewer for spotting this inadequate use of terminology.

However, there is not a clear/explicit justification on how do the chosen characteristics/traits inform us about the actual “sensitivity” and “adaptive capacity” of islands/species to CC. You do provide some of these “justifications” on your methods section but I believe these should come clearly in the introduction. More importantly, given that “adaptive capacity” and “exposure” were the most important components defining islands/archipelagos vulnerability, and that for most “vulnerable archipelagos” the exposure component was not that important, the need to provide an explicit (empirical or at least theoretical, with possibly the latter being more attainable) link between “adaptive capacity”, the specific traits considered and their relationship with climate change is fundamental. Otherwise, it could be argued that all of these traits could have been measured and described to define islands/species vulnerability to any threat without having to rely on CC. That is, you would have found “a clear pattern of spatial heterogeneity in terms of island and archipelago vulnerability” without even considering exposure to CC simply as a result of islands/species intrinsic characteristics.

Moreover, some of these traits are perhaps more relevant to other threats rather than to CC. For example, “protection status” may say nothing on CC adaptability as has been repeatedly suggested (e.g. Hannah 2008; Monzon et al. 2011; Melillo et al. 2016). Similarly, extinction rate (particularly in the way you calculated it, see below) and phylogenetic distinctiveness may be silent regarding CC adaptability. Please, don’t get me wrong, I’m not saying these traits (and the other ones you used) may not be related in some way to CC adaptability or that your approach is not valid, but only that the explicit justification for doing so is lacking from your manuscript.

→ This is a very relevant point and we thank the reviewer for pointing it out. We rewrote some part of the methods to enhance justification of the different variables used to characterize sensitivity and adaptive capacity (L324-343 and 344-381) in relation to CC. We now mentioned in the Introduction that both vulnerability components were characterized by several species and insular characteristics that have been shown to be important response variables to climate change (L83-86). Moreover, we added a supplementary table that recaps the different justifications (Supplementary Table 1).

We also discussed the relevance of some traits (e.g. extinction rate (L197-205), protected areas (L209-213)) and also performed sensitivity analyses to determine how the distribution

patterns of vulnerability component values differ when some variables are removed (see Supplementary Fig. 4 and 5).

Related to the importance of “adaptive capacity” vs exposure, taken at face value, could it then be said that “historical factors” (phylogenetic distinctiveness, extinction rate) may be more important than changing climate alone (exposure: changes in the species’ physical environment) to define vulnerable islands/archipelagos? In any case, such possibility could be considered in your discussion section.

→ We agree with the reviewer and we now discussed this matter (L184-193).

A somewhat less important issue is your “temporal window”. You modeled climate change by 2050, which could be relatively short considering the scale of species responses and threat classification. For instance, to be considered extinct, species must have not been seen by >50 years. In addition, perhaps the adverse effects of CC in the next 30 years may not be observable and thus your “predictions” may be overstated. Again, I’m not saying they are, but that you could justify your choice of time period more convincingly. From what I remember, in the “Global Change” literature and particularly in the “Ecological Niche/Species Distribution Modeling” (ENM/SDM) literature, climate projections consider two or three time windows (e.g. 2050, 2070, 2100) to evaluate potential impacts at different time periods while considering species’ responses at broader time scales.

→ We understand that 2050 could appear very short considering the scale of species responses and threat classification. However, the CR, EN or VU categories of IUCN are established based on a population size reduction of over 90%-70%-50% over the last 10 years or 3 generations. Moreover, many studies showed that uncertainties of climate models output increase over time so by using the closest climate scenarios, we keep the uncertainty linked to global climate models at minimum (Sanderson & Knutti (2012); Thuiller *et al.* (2019)). Also, the effects of climate change, for several decades, are already visible on many characteristics of vertebrates (phenology shifts, species range shifts, changes in body size) and many studies predicting the effects of the coming decades also show significant potential effects. Finally, conservation target (e.g., Aichi target) is usually established at a very short scale (2010 for 2020), thus this timeframe seems appropriate in a conservation perspective. Consequently, and because we agree with the reviewer that is this a less important issue that would not bring important added value to our study, we did not replicate our entire study for an additional time horizon.

References:

Sanderson, B., & Knutti, R. (2012), Climate change projections: Characterizing uncertainty using climate models, in *Climate Change Modeling Methodology*, pp. 235–259, Springer, New York.

Thuiller, W., Guéguen, M., Renaud, R., Karger, D.N., & Zimmermann, N.E. (2019), Uncertainty in ensembles of global biodiversity scenarios. *Nature Communications* 10, 1446.

Minor issues

Introduction

I believe that contextualizing your setting against known patterns in the continents/mainlands would be much informative for the reader. For instance, how “particularly prone” are island species in comparison to continental/mainland species?

→ We have now added a sentence to explain that insular species are more exposed to climate change vulnerability due to the intrinsic and extrinsic properties of insular species (L34-38).

Lines 34-35. 20% refers to island endemics or all species or all endemic species worldwide?

→ We modified information about island endemics to set against known patterns in the mainland (L38-41).

Lines 60-62. To support current validity of this argument, which is certainly true, I think you could cite more current literature. Otherwise, it gives the impression that this was done in the past and we are not sure what is being done nowadays.

→ We now added more recent references about the application of the different methods in the context of climate change (L66-70).

Line 76. "...high vulnerability of this taxa" compare to what?

→ We modified the sentence to avoid implying that we are comparing the "...high vulnerability of this taxa" to something else (L81-82).

Results

Fig. 2b. You could include a vertical line at 0.5 in the vulnerability axis, just for reference.

→ We added a vertical line at 0.5 in the vulnerability axis.

Figs. 2c,d. I wonder if "Exposure" would be better placed at the X-axis, considering that this is an external measure that could be independent of the species whereas the other ones (sensitivity, adaptive capacity) include intrinsic properties of the species. Just an idea.

→ As recommended by the reviewer, we placed "Exposure" at the x-axis.

Lines 105-106. Yes, but by how much? Sure, the details are in SuppTable 3, but could you provide some description on the magnitude of the components' effects?

→ We now added correlation information about variables characterizing the dimensions of the PCA in the main text (L120-124), and also added correlation information between vulnerability and its components (L125-127).

Lines 112-113. All of these characteristics are independent of climate, right?

→ Indeed, geographic isolation (extinction rate or phylogenetic distinctiveness) are not directly related to climate change and the adaptive capacity could also be associated with other threats that are not restricted to climate change (L132-134). However, geographic isolation prevents species to shift in latitude and therefore represent a potential lack of adaptive capacity to climate change. We also argue L363-381, that the two others characteristics are associated with adaptive capacity to climate change.

Discussion

Lines 136-141. But, only two of these were highly "exposed".

→ We added at the beginning of the sentence that only two archipelagos were identified as highly exposed (L164-167).

Line 147. "...high risk of extinction following climate change...", but other factors/traits (intrinsic and perhaps unrelated to climate or indirectly so) determined their "vulnerability" and not only "exposure" per se.

→ We agree with the reviewer that not only exposure per se is important to determine risk of species extinction following climate change. So we reformulated the sentence in order to clarify that risk of species extinction is determined by the climate change vulnerability (L176-177).

Lines 155-158. Agree, but this would be valid only if clearly justify the link between the traits used and climate change “resistance”. See my main concern above.

→ To follow the reviewer’s suggestion, we rewrote some text parts (L83-86; L324-343 and 344-381) and added a supplementary table (Supplementary Table 1) to enhance justification of the different variables used to characterized sensitivity and adaptive capacity.

Line 163. “Species extinction in this region...” which region? West Indies or Japan?

→ We modified the sentence in order to be clearer (L197-199).

Lines 177-180. So, why use it in the first place? Again, the need to provide a stronger justification is fundamental.

→ We have now deleted this sentence which can be misleading, and provided a stronger justification in methods (L363-372; Supplementary Table 1).

Lines 180-181. Thus, a stronger justification on your choices is needed!

→ To do so, we rewrote some parts (L83-86; L324-343 and 344-381) and add a supplementary table (Supplementary Table 1) to enhance justification of the different variables used to characterized sensitivity and adaptive capacity.

Lines 195-196. “...which questions the similarity of responses...for island and continental species...” why? Not clear, please explain.

→ We have now rephrased the sentence to explain why we could observe differences between our result and previous studies (L228-232).

Lines 213-214. “...highlighting archipelagos that were already identified...” For example? Please, repeat them here just for clarity.

→ We repeated the archipelagos who were already identified as hotspots of climate change risk for mammals (L251-254).

Line 227. Just wondering, and based on your previous work, why not include “sea level rise” in your exposure component?

→ Indeed, sea-level rise is an important component of climate change vulnerability. However, mechanisms by which species are vulnerable to sea-level rise is very different than from a change in temperature and/or precipitation regime. Therefore, this would imply to use specific sensitivity and adaptive capacity variables to investigate the vulnerability to sea-level rise. Besides, specific GCMs exploring sea level rise scenarios are unavailable and though our previous works allow us to give a first approximation of the consequences of sea-level rise, we now need more accurate sea-level rise scenario to explore both the effects of temperature and/or precipitation changes and sea-level rise.

Lines 231-232. Again, only two of these are actually “exposed” to CC.

→ We added at the fact that only two archipelagos were identified as highly exposed (L273-274).

Methods

Exposure: how much of this “exposure” would species actually suffer? Any idea? Is there any data/result (e.g. from ENMs/SDMs) that you could use to contextualized (relative to...) the magnitude of “climate change” (based on your measured distance) to the known/expected tolerance of mammal species? Any discussion on this issue, if possible, would be very informative.

→ Our metric of exposure is based on a distance metric to local climatic variables. We chose this metric because it could be standardized among insular species without any prior knowledge about species preferences in terms of climate, habitats, niche, etc ... using SDMs/ENMs would mix exposure and sensitivity components because it relies on species preferences. Therefore, apart from the facts that most endemic insular species suffer from insular syndromes and are more likely to be habitat/diet specialists, which make them particularly vulnerable to climate change, we don't have specific information about tolerance of mammal species. We discussed about it L184-190.

Lines 276-278. In the same vein, how high is “high”? Any reference value for comparison?

→ We have now added information about the local climate change values obtained in the context of our study (L317-318). Nevertheless, the few studies applying this kind of analysis did not provide quantitative and accurate information on the values of local climate change, making it difficult to compare our values with other studies.

Lines 289-296. Some of this could be in the intro (justification).

→ We now mentioned in the Introduction that both sensitivity and adaptive capacity components were characterized by several species and insular characteristics that have been shown to be important response variables to climate change (L83-86); and we added a supplementary table that recap the different justifications (Supplementary Table 1).

Adaptive capacity: Why this phylogeny? Any justification based on your aims? What about using Bininda-Emonds *et al.* (2007 *Nature*; Fritz *et al.* 2009 *EcolLett*) phylogeny, Hedges *et al.* (2015 *MolBiolEvol*) or even Upham *et al.* (2019 *PLoS Biol*)? This is particularly relevant given that your metric depends on phylogenetic distances/branch lengths that were particularly highlighted by the original authors (Faurby & Svenning) as problematic: “...analyses using the resulting phylogeny should focus on the topology rather than on branch lengths” (pg. 16 in Faurby & Svenning 2015).

→ Indeed, the authors of these studies advise to focus on the topology rather than on branch lengths by using their mammals' phylogeny, but these authors also used their phylogeny to examine the loss of evolutionary history across mammals (based on branch lengths; Davis *et al.* 2018). The authors highlighted that by using their phylogeny, “this is unlikely to have a major effect on their results though; global estimates of PD in mammals are remarkably robust to phylogenetic inaccuracies (Rodrigues *et al.* 2011)”. In addition, to account for phylogenetic uncertainty, the authors performed PD analyses on a randomly subset of 30 trees on the 1,000 available with randomly resolved polytomies, and showed that values obtained from the trees subset are nearly identical showing that the number, although arbitrary, is sufficient to accurately capture phylogenetic uncertainty in their analyses.

In our case, as macro-ecology studies using phylogeny data used only one source, we choose to use the most recent phylogeny when we performed analyses (here PHYLACINE from Faurby, S. *et al.* 2018), and used the 1,000 trees available with randomly resolved polytomies to account for phylogenetic uncertainty. We now added this information L367-369.

References:

Davis *et al.* (2018) Mammals diversity will take millions of years to recover from the current biodiversity crisis. *PNAS* 115, 11262–11267.

Faurby, S. *et al.* (2018) PHYLOCINE 1.2: The Phylogenetic Atlas of Mammal Macroecology. *Ecology* 99, 2626–2626.

Rodrigues ASL, *et al.* (2011) Complete, accurate, mammalian phylogenies aid conservation planning, but not much. *Philos Trans R Soc London, Ser B* 366, 2652–2660.

Line 323. What is the meaning of “resistance capacity”? Please, explain.

→ We reformulated the sentence in order to be clearer about what we meant by “resistance capacity”. Specifically, we deleted “resistance capacity” term and replaced it by “more phylogenetically diverse species pools may have a higher evolutionary potential to adapt” (L369-372).

Lines 323-328. Extinction rate: how representative it is regarding your dataset? Why not considered only mammals instead of vertebrates?

→ If no mammal extinction has taken place on an island, we cannot disentangle if this is due to the absence of perturbations on the islands or because mammals were “resistant” to past perturbations, especially because on the 340 islands, only 26 were the place of extinction of mammals. The number of islands up to 88 when all vertebrate species are considered. In addition, we found a fair correlation between mammal and vertebrate extinctions (0.5).

Lines 328-331. How? Please, explain. Did you obtain a species-level (then averaged across species) or an island-level value?

→ The extinction rate is calculated by the ratio between the number of vertebrate extinctions since the year 1500 CE and the total species richness of vertebrates for each island, and so is an island-level value (L373-374).

References:

Monzón, J., Moyer-Horner, L., & Palamar, M. B. (2011). Climate change and species range dynamics in protected areas. *Bioscience*, 61(10), 752-761.

Hannah, L. (2008). Protected areas and climate change. *Annals of the New York Academy of Sciences*, 1134(1), 201-212.

Melillo, J. M., Lu, X., Kicklighter, D. W., Reilly, J. M., Cai, Y., & Sokolov, A. P. (2016). Protected areas’ role in climate-change mitigation. *Ambio*, 45(2), 133-145.

REVIEWERS' COMMENTS:

Reviewer #1 (Remarks to the Author):

I appreciate the thoughtful response to my previous review. I believe the authors have fully addressed my concerns and I have no additional concerns at this time.

Reviewer #2 (Remarks to the Author):

I believe the authors have done a good job exploring the sensitivity of their vulnerability assessment.

The fact that some spatial patterns changed is actually an interesting results, since it shows that some criteria may make some island more vulnerable than others.

I believe the ms is ready for publication in Nature Communications

Reviewer #3 (Remarks to the Author):

Dear colleagues,

Thank you for considering and addressing all of my previous criticisms. I am satisfied by your responses and corresponding changes to your Ms. I was particularly pleased by the addition of supplementary table 1, which deals with the justification of your chosen variables, and the sensitivity analyses as a response to other reviewer's suggestions (e.g. Figs. S4 and S8). If the journal allows it, perhaps such table S1 could be included in the main text to strengthen your argument. Just a suggestion if space allows it. Finally, I would only suggest to carefully check that your new text/changes are in the appropriate places since I noticed some inconsistencies in the line numbers between your rebuttal letter and the main text.

I hope my comments were useful and constructive.

Best,

REVIEWERS' COMMENTS:

Reviewer #1 (Remarks to the Author):

I appreciate the thoughtful response to my previous review. I believe the authors have fully addressed my concerns and I have no additional concerns at this time.

→ We thank the reviewer for her/his positive appreciation on the revision of our manuscript.

Reviewer #2 (Remarks to the Author):

I believe the authors have done a good job exploring the sensitivity of their vulnerability assessment.

The fact that some spatial patterns changed is actually an interesting results, since it shows that some criteria may make some island more vulnerable than others.

I believe the ms is ready for publication in Nature Communications

→ We thank the reviewer for her/his positive appreciation on the revision of our manuscript.

Reviewer #3 (Remarks to the Author):

Dear colleagues,

Thank you for considering and addressing all of my previous criticisms. I am satisfied by your responses and corresponding changes to your Ms. I was particularly pleased by the addition of supplementary table 1, which deals with the justification of your chosen variables, and the sensitivity analyses as a response to other reviewer's suggestions (e.g. Figs. S4 and S8). If the journal allows it, perhaps such table S1 could be included in the main text to strengthen your argument. Just a suggestion if space allows it. Finally, I would only suggest to carefully check that your new text/changes are in the appropriate places since I noticed some inconsistencies in the line numbers between your rebuttal letter and the main text.

I hope my comments were useful and constructive.

Best,

Fabricio Villalobos

→ We thank the reviewer for his positive appreciation on the revision of our manuscript. As Nature Communications articles may have up to 10 display items, we moved the supplementary table 1 in our main text. Now, our manuscript has four figures and one table.

It seems that when our word manuscript was converted to pdf on the Manuscript Tracking System, the line numbers have changed. We should have checked this before submitting our manuscript and we apologize for any inconvenience caused.